# Rapid Analysis of Aristolochic Acids and Aristolactams in Houttuyniae Herba by LC–MS/MS

**DOI:** 10.3390/molecules27248969

**Published:** 2022-12-16

**Authors:** Yingxue Wu, Jing Liu, Shuai Kang, Zhong Dai, Shuangcheng Ma

**Affiliations:** Institute for Control of Chinese Traditional Medicine and Ethnic Medicine, National Institutes for Food and Drug Control, Beijing 100050, China

**Keywords:** LC–MS/MS, aristolactams, aristolochic acids, Houttuyniae herba, alkaloids

## Abstract

Houttuyniae herba, the Saururaceae plant *Houttuynia cordata* Thunb., has multiple therapeutic effects, including clearing heat, eliminating toxins, reducing swelling, discharging pus, and relieving stagnation. It has a long history as an edible and medicinal plant in China. Phytochemical studies show that the main constituents include volatile oil, flavonoids, and alkaloids. Aristolactam is a major alkaloid with a structure similar to toxic aristolochic acids. However, there has been no systematic study on aristolochic acids and alkaloids in Houttuyniae herba. Therefore, in this study, an LC–MS/MS method was developed to simultaneously detect seven alkaloids and five aristolochic acids in Houttuyniae herba from different origins. Six alkaloids (O-demethyl nornucifrine, N-nornucifrine, aristololactam AII, aristololactam FI, aristololactam BII, cepharadione B) were found and quantitatively determined in 75 batches of samples. Meanwhile, no aristolochic acids or aristololactams were found in Houttuyniae herba at a limit of detection (LOD) of ≤4 ng/mL. The method developed was fully validated in terms of LOD, limit of quantification (LOQ), linearity, precision, accuracy, and stability. These data clarify the content of the above safety-related components in Houttuyniae herba and provide a reference for further research into its safety.

## 1. Introduction

*Houttuynia cordata* Thunb. (Saururaceae) is a medicinal and food dual-purpose plant in Asia. Its fresh underground stems have good dietary value and therapeutic benefits. Additionally, its whole fresh plant or the dried aerial part is also used extensively as herbal medicine in treating various diseases [1,2,3]. It possesses the actions of clearing heat, eliminating toxins, reducing swelling, discharging pus, and relieving stagnation. Modern studies revealed that volatile oils, flavonoids, and alkaloids are the main components. These components have several bioactivities, including anti-inflammation, antiviral, antitumor, immune modulation, antioxidation, and antibacterial [4,5,6,7,8,9].

Literature research shows that 11 aristololactams (aristololactam AII, aristololactam FII, aristololactam BII, aristololactam FI, piperolactam B, piperolactam C, piperolactam D, aristololactam FI, 3-hydroxy-1,2-dimethoxy-5-methyl-5H-dibenzoindol-4-one, 3-methoxy-5-methyl-5H-benzodioxolo-benzoindol-4-one, and 3,4-dimethoxy-N-methyl aristolactam) have been isolated from Houttuyniae herba [10,11,12]. Aristolactams are naturally occurring phenanthrene lactam alkaloids, which are the main metabolites of aristolochic acid by nitro-reduction reaction in vivo. It has been proved that some aristolochic acids have renal toxicity, cause carcinogenesis, and may cause gene mutations [13,14,15,16]. The use of aristolochic acid-containing herbal medicines is forbidden in many countries. However, there are insufficient studies on the safety and quantitative analysis of aristololactam components. It is necessary to develop a method for identifying and quantifying of aristolochic acids and alkaloids in Houttuyniae herba.

Liquid chromatography–tandem mass spectrometry (LC–MS/MS) was used to analyze Houttuyniae herba samples for the natural existence of aristolochic acid I (AA-I) and aristolochic acid II (AA-II) [17]. Studies revealed that neither AA-I nor AA-II exist naturally in Houttuyniae herba or are below the method detection limits (MDLs; <2 ng/g). In this study, LC–MS/MS was used to detect 12 compounds in Houttuyniae herba from different origins, wild and cultivated, to provide a reference for the research into the safety of Houttuyniae herba.

## 2. Results

An established LC–MS/MS method for seven alkaloids and five aristolochic acids was applied to analyze Houttuyniae herba, and six alkaloids [O-demethyl nornucifrine (C1), N-nornucifrine (C2), aristololactam AII (C5), aristolactam FI (C7), aristolactam BII (C9), and cepharadione B (C11)] were identified. The typical MRM chromatograms for a mixed standard solution and a sample of Houttuyniae herba are shown in Figure 1. These data showed that the method is highly selective.

### 2.1. Linearity Range, Limits of Detection (LODs), and Limits of Quantification (LOQs)

Working standard solutions containing seven alkaloids and five aristolochic acids were prepared by series dilution of the mixed stock solution with 80% methanol to different concentrations. Then, they were injected and analyzed. The regression equations, linearity, determination coefficient, and limits of detection and quantification of the method are presented in Table 1. All calibration curves showed good linear regression (R^2^ ≥ 0.9911) within the tested ranges. We precisely diluted the stock mixed solution with methanol quantitatively and stepwise if necessary. The diluted solutions were separately injected and analyzed. The limit of detection (LOD) and limit of quantification (LOQ) (Table 1) were defined as the concentrations that could be detected and yield signal-to-noise (S/N) ratios of 3:1 and 10:1, respectively, according to guidelines for validation of analytical methods for pharmaceutical quality standards.

### 2.2. Precision

The precision of the method was evaluated based on intra- and inter-day precision. The intra-day precision was tested with mixed standard solutions over 1 day. The standard solutions were examined in triplicate on three consecutive days for inter-day precision. The corresponding % RSD values were calculated. The RSDs for the intra-day (n = 6) and inter-day (n = 9) assays were less than 3.5% and 4.8%, respectively (see Table 2).

### 2.3. Stability and Repeatability

The stability was measured using a sample solution (S14) and performed at 0, 2, 4, 8, 12, and 24 h after preparation and storage at room temperature. Six independent sample solutions were prepared and analyzed to measure the repeatability. The concentration of each solution was determined by calibration curves produced on the same day. The RSDs for stability were less than 5.6% within 24 h. Moreover, the RSDs for repeatability were less than 5.8% (Table 2). The stability and repeatability tests show that all analytes are stable within the whole analysis and that the test method is sufficiently effective for conventional analysis.

### 2.4. Recovery

The recovery experiment was performed by adding a known amount of individual reference standards into a certain amount of sample (S14). Nine replicates were performed for the test.

The recoveries were calculated using the following equation: recovery (%) = (total amount detected—amount original)/amount spiked × 100%. The results show that the average recoveries ranged from 77% to 120% with RSDs in the range of 1.0–5.8%, indicating that the method was accurate (see Appendix A).

### 2.5. Sample Analysis

Seventy-five batches of samples were prepared and analyzed according to 2.5, 2.6, and 2.7, and the quantification results are summarized. Compounds aristolochicacid IIIa (C3), 7-hydroxy aristolochic acid I (C4), aristolochic acid Iva (C6), aristolochic acid II (C8), aristolactam I (C10), and aristolochic acid I (C12) were not detected in the 75 batches of samples.

The results showed that the content of compounds in dried Houttuyniae herba (Figure 2) followed the order: O-demethyl nornucifrine (C1) < N-nornucifrine (C2) < aristololactam AII (C5) < aristolactam FI (C7) < cepharadioneB (C11) < aristolactam BII (C9). The total content of the six alkaloids in dried Houttuyniae herba was as follows: aerial part (31–530 μg/g), leaves (12–870 μg/g), and aerial stem (12–380 μg/g). The average content of the total content was: aerial stems (72 μg/g) < leaves (180 μg/g) (see Appendix A).

The total content of six alkaloids in fresh Houttuyniae herba (Figure 3) (calculated as dry) was as follows: underground stem (6.5–19 μg/g) and whole grass (110–130 μg/g), respectively. The underground stem has a lower potential risk (see Appendix A).

## 3. Discussion

### 3.1. Optimization of the Extraction Method Optimization of MS Conditions

A sample (S14) was used to optimize the extraction process. Optimization was completed using a three-step approach, which can be described as follows. Step 1. Optimization of the extraction solvent system: the first step in preparing the sample solution was to select a suitable extraction solvent because of its paramount role in achieving good recovery. Six solutions [50%, 60%, 70%, 80%, 90%, and 100% methanol (*v*/*v* in water)] were systematically compared considering the peak areas of the six alkaloids in Houttuyniae herba. The result was that 80% methanol exhibited the highest extraction efficiency among the tested solvents (Figure 4a). Hence, 80% methanol was selected as the best extraction solvent for this study. Step 2. Optimization of solvent volume: extractant volume may have been another factor to affect extraction efficiency. This study aimed to obtain the minimum volume of extractant required to achieve the highest extraction efficiency. Five different volumes of methanol (10, 20, 30, 40, and 50 mL) were systematically studied. The peak areas of the six alkaloids increased with an increasing volume of methanol (Figure 4b). However, there was no significant difference among the results of five different volumes of methanol. Therefore, 20 mL was eventually selected as the optimized volume for environmentally friendly reasons. Step 3. Optimization of ultrasonication time: in this study, an ultrasonic process was used to extract the six alkaloids from Houttuyniae herba. There was no significant difference among ultrasonication times of 10, 20, 30, 40, and 50 min (Figure 4c). Accordingly, 30 min was selected as the best extraction time to save energy.

In conclusion, the optimal sample preparation method was extracting of a 0.5 g sample with 20 mL of 80% methanol in an ultrasonic water bath for 30 min. 

### 3.2. Optimization of LC–MS/MS Conditions

The chromatographic conditions, especially the mobile phase composition, were optimized to achieve the best possible resolution and symmetric peaks of the six compounds within a suitable run time. Throughout the tests, three mobile phases were examined: acetonitrile–water, acetonitrile–0.1% formic acid, and acetonitrile–0.1% formic acid (containing 5 mM ammonium acetate). The acetonitrile–water containing 0.1% formic acid (*v*/*v*) combination had the lowest pressure, best baseline stability, and highest ionization efficiency among those tested and was eventually selected as the mobile phase [18].

## 4. Materials and Methods

### 4.1. Chemicals and Reagents

O-Demethyl nornucifrine (C1) (HPLC purity ≥99.74%, Lot no. ZT-24107) and N-Nornucifrine (C2) (HPLC purity ≥99.82%, Lot no. ZC-53802) were from Shanghai Zhenzhun Biochemical Co., Ltd (Shanghai, China). Aristolochic acid IIIa (C3) (HPLC purity ≥99%, Lot no. C11188894) and 7-hydroxy aristolochic acid I (C4) (HPLC purity ≥98%, Lot No. C12347780) were from Shanghai Macklin Biochemical Co., Ltd (Shanghai, China). Aristololactam AII (C5) (HPLC purity ≥97%, Lot no. X27S11L125977), aristololactam FI (C7) (HPLC purity ≥98%, Lot no. X09M11L112632), aristolochic acid II (C8) (HPLC purity ≥98%, Lot no. P13J10F90613), aristololactam BII (C9) (HPLC purity ≥97%, Lot no. X09M11L112631), and aristololactam I (C10) (HPLC purity ≥98%, Lot no. P27N10S104067) were from Shanghai Yuanye Bio-Technology Co., Ltd (Shanghai, China). Aristolochic acid IVa (C6) (HPLC purity ≥98%, Lot no. DST190415-057) was from Chendu DeSiTe Bio-Technology Co., Ltd (Chendu, China). Cepharadione B (C11) (HPLC purity ≥95%) was prepared in the laboratory. Aristolochic acid I (C12) (HPLC purity ≥99.1%, Lot no. 110746–201912) was from National Institutes for Food and Drug Control, Beijing, China (Figure 5). Methanol (analytical reagent) was from National Drug Chemical Reagents Co., Ltd (Beijing, China). Acetonitrile (mass spectrometry reagent) and formic acid (mass spectrometry reagent) were from Thermo Fisher Scientific (Waltham, MA, USA). Water was of ultrahigh purity.

### 4.2. Samples

Seventy-five batches of Houttuyniae herba samples were collected from different provinces in China (Table 3). It should be noted that the leaves (S32–50) and aboveground stems (S54–72) come from different parts of aboveground parts (S1–15 and S17–20). The leaves (S51–53) and aboveground stems (S73–75) come from different parts of the aboveground part (S21–23).

The samples were authenticated by Associate Professor Shuai Kang (Institute for Control of Chinese Traditional Medicine and Ethnic Medicine, National Institutes for Food and Drug Control, National Medical Products Administration (Beijing, China)).

### 4.3. Instrumentation

A Shimadzu LC–MS/MS 8050 (Shimadzu Co., Kyoto, Japan) equipped with an electrospray ionization device was used for sample analysis. We also used a METTLER XS105 electronic analytical balance (Mettler-Toledo, Zurich, Switzerland), Milli-Q water purification system (Millipore, Burlington, NJ, USA), and KQ-500DE numerical control ultrasound cleaning instrument (Kun Shan Ultrasonic Instruments Co., Ltd., Kunshan, China).

### 4.4. Preparation of Standard Solutions

Standard stock solutions of O-demethyl nornucifrine (C1), N-nornucifrine (C2), aristolochic acid IIIa (C3), 7-hydroxy aristolochic acid I (C4), aristololactam AII (C5), aristolochic acid Iva (C6), aristolactam FI (C7), aristolochic acid II (C8), aristolactam BII (C9), aristolactam I (C10), cepharadione B (C11), and aristolochic acid I (C12) were prepared by dissolving suitable amounts of reference substance in 80% methanol (*v*/*v* in water) to make the concentration of 80 μg/mL. The mixed standard stock solution was freshly prepared by combining an appropriate amount of each standard stock solution and diluting it with a known volume of 80% methanol.

### 4.5. Sample Preparation

For Houttuyniae herba pulverized to powder, we weighed 0.5 g samples accurately and placed them into a 50 mL plug conical bottle. Twenty milliliters of 80% methanol were added precisely and weighed, respectively. After ultrasonic extraction (power: 500 W; frequency: 40 kHz) for 30 min, the extract was cooled down and then the lost weight was made up by adding 80% methanol. This extract was then filtered through a 0.22 μm microporous filter membrane. For each of the 18 batches of fresh Houttuyniae herba (S16, S21–31, S51–53, S73–75), after drying, we performed the same treatment as described for the already-dried products.

### 4.6. HPLC Chromatographic Conditions, Instruments, and Analytical Conditions

Using a gradient elution, the chromatographic separation was achieved on an Agilent SB-C18 (2.1 × 50 mm, 1.8 μm) at 30 °C. The mobile phase consisted of solution A (0.1% formic acid in water) and solution B (acetonitrile). The gradient elution profile was as follows: 0–10 min, 25% B; 10–12 min, 25–40% B; 12–17 min, 40% B; 17–17.01 min, 40–80% B; 17.01–20 min, 80% B; 20–20.01 min, 80–25% B; 20.01–25 min, 25% B. The flow rate was 0.3 mL/min, the autosampler temperature was 10 °C, and the sample volume injected was 1 μL. Before sample analysis, the column was equilibrated with the mobile phase at 25% B for 30 min.

### 4.7. MS Conditions

The triple-quadrupole MS equipped with a positive electrospray ionization source was used in the MRM mode. The electrospray ionization mass spectrometry (ESI–MS) parameters were as follows: interface temperature, 300 °C; desolvation line (DL) temperature, 250 °C; heat block temperature, 400 °C; nebulizer gas flow rate, 3 L/min; heating gas flow rate, 10 L/min; and drying gas flow rate, 9 L/min.

The MRM conditions were individually optimized for each of the 12 compound reference standards because of their different structures. The MS conditions for MRM are summarized in Table 4, and the typical MRM chromatogram is shown in Figure 1.

## 5. Conclusions

In this study, an LC–MS/MS method for qualitative and quantitative detection of seven alkaloids (O-demethyl nornucifrine, N-nornucifrine, aristololactam AII, aristololactam FI, aristololactam BII, aristololactam I, and cepharadione B) and five aristolochic acids (aristolochic acid IIIa, 7-hydroxy aristolochic acid I, aristolochic acid IVa, aristolochic acid II, and aristolochic acid I) was established. This method has the outstanding advantages of strong specificity, high sensitivity, high accuracy, good reproducibility, and high throughput automation. The content of the above compounds in dried and fresh samples of Houttuyniae herba was detected for the first time. The safety was not related to aristolochic acid IIIa, 7-hydroxy aristolochic acid I, aristolochic acid IVa, aristolochic acid II, aristolochic acid I, or aristololactam I. Therefore, a follow-up study on the safety of alkaloids in Houttuyniae herba should be a focus. The content of the six alkaloids in aerial stems was less than in leaves. Our findings suggest the edible underground stem has a relatively lower potential risk than aerial plant parts. This study is of great significance for the safety evaluation of Houttuyniae herba. It also provides a scientific basis for follow-up safety risk control measures.

## Figures and Tables

**Figure 1 molecules-27-08969-f001:**
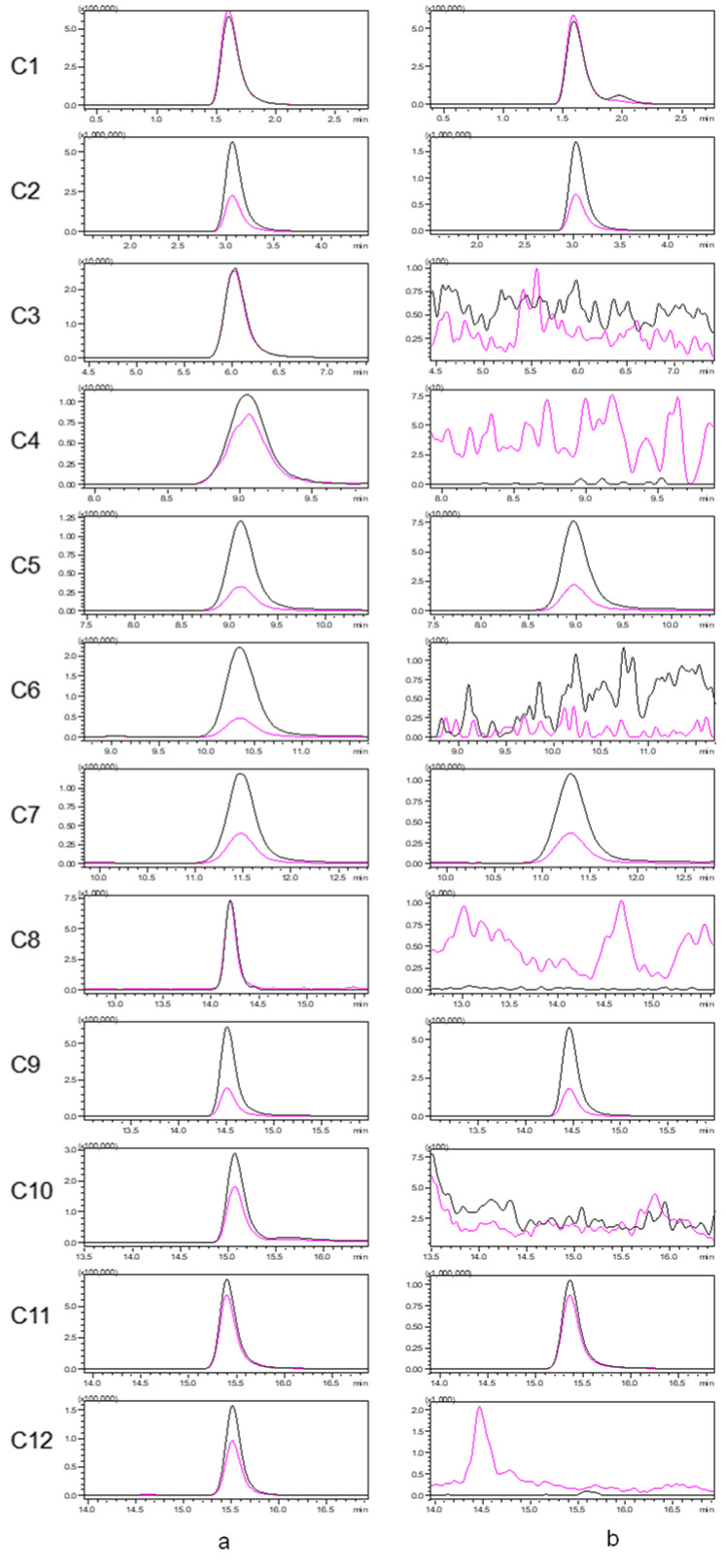
Typical multiple reaction monitoring (MRM) chromatograms for a mixed standard solution (**a**) and a sample of Houttuyniae herba (**b**). [The quantitative ion pairs: C1(*m*/*z* 282.0 > 251.1); C2(*m*/*z* 282.0 > 265.1); C3(*m*/*z* 344.9 > 282.0); C4(*m*/*z* 374.9 > 314.1); C5(*m*/*z* 266.0 > 251.1); C6(*m*/*z* 374.9 > 312.0); C7(*m*/*z* 266.0 > 251.1); C8(*m*/*z* 329.0 > 268.1); C9(*m*/*z* 279.9 > 264.1); C10(*m*/*z* 294.0 > 278.9); C11(*m*/*z* 322.0 > 306.0); C12(*m*/*z* 359.0 > 298.1)].

**Figure 2 molecules-27-08969-f002:**
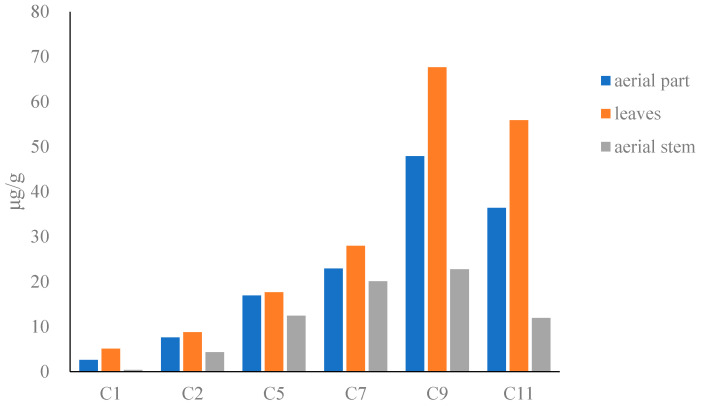
Bar graph of content of six alkaloids in dried Houttuyniae herba.

**Figure 3 molecules-27-08969-f003:**
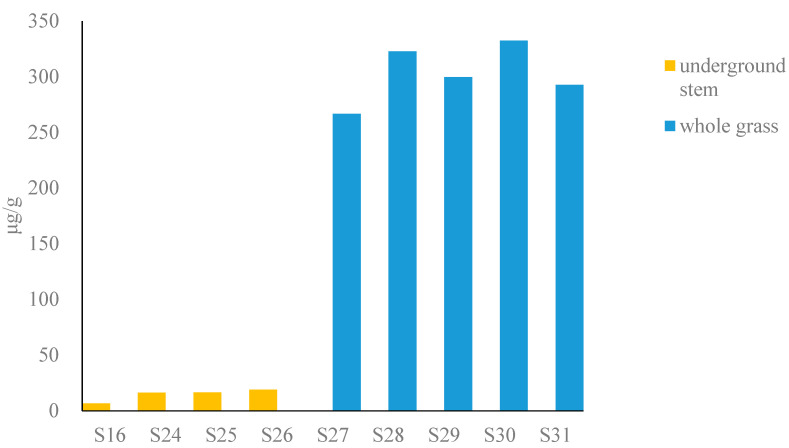
Bar graph of content of total alkaloids in fresh Houttuyniae herba.

**Figure 4 molecules-27-08969-f004:**
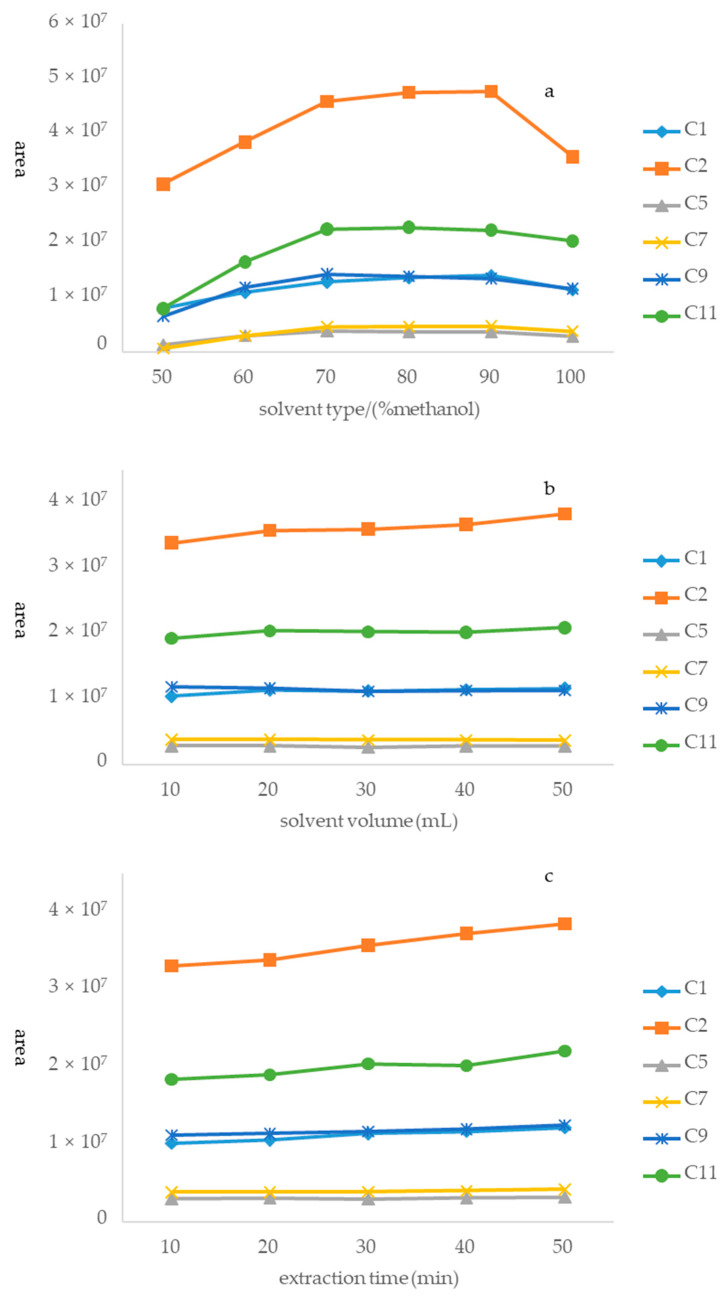
Optimization of different parameters of the method of sample solution: (**a**) type of extractant, (**b**) volume of extractant, and (**c**) ultrasound time.

**Figure 5 molecules-27-08969-f005:**
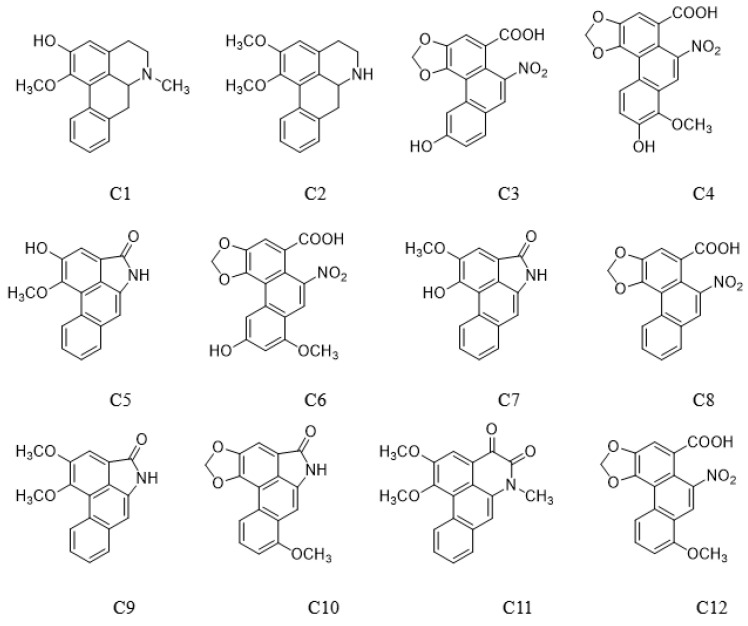
Chemical structures of 12 compounds.

**Table 1 molecules-27-08969-t001:** Regression equation, LOD, and LOQ of the six dianthrones of 12 compounds.

	Regression Equation	Linear Range (ng/mL)	R^2^	LOD (ng/mL)	LOQ (ng/mL)
C1	y = 74,817.72 × x + 203.01	1.8~88	0.9999	0.035	0.116
C2	y = 85,419.58 × x + 589.94	15~760	0.9998	0.03	0.099
C3	y = 10,317.74 × x + 278.40	400~8000	0.9986	3.992	13.174
C4	y = 17.13 × x − 13349.20	270~5400	0.9997	2.684	8.857
C5	y = 184.43 × x + 170.21	18~910	0.9996	0.365	1.205
C6	y = 7102.95 × x + 184.16	270~5400	0.9994	2.687	8.867
C7	y = 141.62 × x + 69.29	22~890	0.9998	0.445	1.469
C8	y = 17,187.75 × x + 158.01	990~9900	0.9911	3.945	13.019
C9	y = 4601.52 × x + 354.20	80~4000	0.9999	0.402	1.327
C10	y = 18,862 × x + 342.51	41~830	0.9994	0.166	0.548
C11	y = 3453.48 × x + 124.20	42~1700	0.9995	0.166	0.548
C12	y = 1265.59 × x − 3289.37	260~5100	0.9994	2.569	8.478

**Table 2 molecules-27-08969-t002:** Stability, repeatability, and precision of analysis of 12 compounds.

	Repeatability		Stability	Precision RSD%	
	Average Content (μg/g)	RSD%	1-2-4-8-12-24h (n = 6)	Intra-Day (n = 6)	Inter-Day( n = 9)
C1	2.34	2.8	2.2	3.5	3.0
C2	7.48	2.3	1.9	2.4	2.4
C3	74.3	4.8	4.5	3.4	4.8
C4	50.9	4.6	3.4	3.3	3.0
C5	9.91	3.9	2.0	3.0	2.9
C6	57.7	2.4	1.5	2.0	1.1
C7	6.49	5.8	1.3	1.3	2.5
C8	111	2.6	3.9	1.6	3.0
C9	32.9	3.3	2.2	2.25	1.1
C10	7.60	1.4	2.4	2.0	1.8
C11	17.9	3.5	5.6	2.3	4.7
C12	38.5	1.4	3.4	2.1	2.5

**Table 3 molecules-27-08969-t003:** Sample collection information for the present study.

Sample Number	Location	Parts	Sample Number	Location	Parts
S1	Yibin, Sichuan Province, China	aerial part	S39	Guanghan, Sichuan Province, China	leaves
S2	Enshi, Hubei Province, China	aerial part	S40	Qionglai, Sichuan Province, China	leaves
S3	Henan Province, China	aerial part	S41	Sichuan Province, China	leaves
S4	Shiyan, Hubei Province, China	aerial part	S42	Sichuan Province, China	leaves
S5	Kaili, Guizhou Province, China	aerial part	S43	Chongqing, China	leaves
S6	Leshan, Sichuan Province, China	aerial part	S44	Hubei Province, China	leaves
S7	Shifang, Sichuan Province, China	aerial part	S45	Sichuan Province, China	leaves
S8	Guanghan, Sichuan Province, China	aerial part	S46	Sichuan Province, China	leaves
S9	Qionglai, Sichuan Province, China	aerial part	S47	Qingtian, Zhejiang Province, China	leaves
S10	Sichuan Province, China	aerial part	S48	Qingtian, Zhejiang Province, China	leaves
S11	Sichuan Province, China	aerial part	S49	Hangzhou, Zhejiang Province, China	leaves
S12	Chongqing, China	aerial part	S50	Yunnan Province, China	leaves
S13	Hubei Province, China	aerial part	S51	Yaan, Sichuan Province, China	leaves (fresh)
S14	Sichuan Province, China	aerial part	S52	Yaan, Sichuan Province, China	leaves (fresh)
S15	Sichuan Province, China	aerial part	S53	Yaan, Sichuan Province, China	leaves (fresh)
S16	Beijing, China	underground stem (fresh)	S54	Yibin, Sichuan Province, China	aerial stem
S17	Qingtian, Zhejiang Province, China	aerial part	S55	Enshi, Hubei Province, China	aerial stem
S18	Qingtian, Zhejiang Province, China	aerial part	S56	Henan Province, China	aerial stem
S19	Hangzhou, Zhejiang Province, China	aerial part	S57	Shiyan, Hubei Province, China	aerial stem
S20	Yunnan Province, China	aerial part	S58	Kaili, Guizhou Province, China	aerial stem
S21	Yaan, Sichuan Province, China	aerial part (fresh)	S59	Leshan, Sichuan Province, China	aerial stem
S22	Yaan, Sichuan Province, China	aerial part(fresh)	S60	Shifang, Sichuan Province, China	aerial stem
S23	Yaan, Sichuan Province, China	aerial part (fresh)	S61	Guanghan, Sichuan Province, China	aerial stem
S24	Zunyi, Guizhou Province, China	underground stem (fresh)	S62	Qionglai, Sichuan Province, China	aerial stem
S25	Guiyang, Guizhou Province, China	underground stem (fresh)	S63	Sichuan Province, China	aerial stem
S26	Bijie, Guizhou Province, China	underground stem (fresh)	S64	Sichuan Province, China	aerial stem
S27	Shifang, Sichuan Province, China	whole plant (fresh)	S65	Chongqing, China	aerial stem
S28	Shifang, Sichuan Province, China	whole plant (fresh)	S66	Hubei Province, China	aerial stem
S29	Shifang, Sichuan Province, China	whole plant (fresh)	S67	Sichuan Province, China	aerial stem
S30	Shifang, Sichuan Province, China	whole plant (fresh)	S68	Sichuan Province, China	aerial stem
S31	Shifang, Sichuan Province, China	whole plant (fresh)	S69	Qingtian, Zhejiang Province, China	aerial stem
S32	Yibin, Sichuan Province, China	leaves	S70	Qingtian, Zhejiang Province, China	aerial stem
S33	Enshi, Hubei Province, China	leaves	S71	Hangzhou, Zhejiang Province, China	aerial stem
S34	Henan Province, China	leaves	S72	Yunnan Province, China	aerial stem
S35	Shiyan, Hubei Province, China	leaves	S73	Yaan, Sichuan Province, China	aerial stem (fresh)
S36	Kaili, Guizhou Province, China	leaves	S74	Yaan, Sichuan Province, China	aerial stem (fresh)
S37	Leshan, Sichuan Province, China	leaves	S75	Yaan, Sichuan Province, China	aerial stem (fresh)
S38	Shifang, Sichuan Province, China	leaves			

**Table 4 molecules-27-08969-t004:** Mass spectrometry parameters.

	Retention Time (min)	Precursor Ion (*m*/*z*)	Product Ion (*m*/*z*)	Q1 Pre-Rod Bias (V)	CE (eV)	Q3 Pre-Rod Bias (V)
C1	1.59	282.0	251.1 *	−14	−17	−29
			219.1	−14	−19	−26
C2	3.00	282.0	265.1 *	−21	−15	−20
			250.1	−15	−21	−29
C3	5.94	344.9	282.0 *	−28	−15	−12
			283.9	−27	−14	−21
C4	8.92	374.9	314.1 *	−18	−13	−17
			340.1	−30	−15	−19
C5	8.97	266.0	251.1 *	−28	−23	−29
			195.1	−28	−31	−23
C6	10.21	374.9	312.0 *	−29	−18	−24
			297.0	−30	−32	−23
C7	11.42	266.0	251.1 *	−28	−23	−29
			195.1	−28	−31	−23
C8	14.17	329.0	268.1 *	−16	−11	−21
			294.0	−14	−16	−22
C9	14.50	279.9	264.1 *	−23	−26	−30
			236.1	−23	−28	−18
C10	14.99	294.0	278.9 *	−25	−29	−20
			251.1	−12	−35	−14
C11	15.40	322.0	306.0 *	−17	−29	−23
			278.1	−17	−29	−21
C12	15.45	359.0	298.1 *	−14	−12	−16
			296.1	−14	−15	−22

* The quantitative ions.

## Data Availability

All data included in this study are available upon request by contact with the corresponding author.

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
