# Peer review of "Rapid Analysis of Aristolochic Acids and Aristolactams in Houttuyniae Herba by LC–MS/MS"

_molecules, 2022, doi:10.3390/molecules27248969_

Round 1
Reviewer 1 Report
The article is poorly written and is very confusing. It definitely does not reflect the quality of the journal. English needs to be overhauled deeply. Much work still needs to be done before its publication can be considered.
Just a few examples of inconsistencies:
- the list of 12 compounds must be removed from the Introduction section.
-Table 4: should be placed in the Supplemental Materials.
-Table 7: I do not find the MS parameters indicated in the caption but data relative to the identification of compounds that should be present in the Results section.
-par.4.6: gradient. Isn't there a better method to indicate the constant composition of the solvent ? 25-25% B etc. ? This is something I have never seen before.
-lines 162-165: ???????????????
And many, many other problems.
Reviewer 2 Report
Like a review
to the manuscript of Wu YingXue and co-authors “Rapid analysis if aristolochic acids and aristolactams in hottuyniae herba by LC-MS/MS”
The manuscript includes important information on the analysis of rather “exotic” type of alkaloids and, hence, deserves attention. The structure of the text (rubricating) seems to be standard for such manuscripts. There are no principal objections to the results.
However, there is one dangerous (in the reviewer’s opinion) problem: it is the number of significant digits in all the numerical data, or, in other words, the necessity of correct rounding. Please look at the pdf-file attached (I will try to mail it to Editorial Office); all the points required corrections are marked just in the text. The number of significant digits should be reduced at the pages 4, 5 – 7 (Tables 2 and 3), 8-9 (Table 4), and in some lines within the text. Really, let us look at the one number from Table 4, namely 58.163±0.554. If the first digit of standard deviation is 5, there are no reasons to indicate all other digits. Finally, the correct presentation should be 58.2 +- 0.6.
The reviewer strongly recommends to Authors to reduce the number of useless digits in most of presented values. After that the manuscript can be recommended for publication in “Molecules”.

Reviewer 3 Report
The report seems important in addressing the difficulties we face in simultaneous determination of such compounds. However, there are various language, spelling, and technical issues to be addressed to enhance the quality of this job.
1. In the abstract, the LOD and LOQ values were expressed as "pg". What does this mean? Please consider this and provide the exact and accurate units of LOD and LOQ.
2. While testing the linearity of the method, what is the basis for selecting concentration range (linearity range)? and why different concentrations were considered for different analytes (e.g. 1.8 - 87.5 for C1 and 986.3 - 9862.7 for C3?
3. Table 1: What are the units for LOD and LOQ concentrations? pg is not a concentration unit as far as I know.
4. To check precision of the method (Line 79 - 83), the authors indicated that they have used mixed standard. What does this mean? Did you spike the mixed standard on the plant extract, extracted it again and did the analysis or just you injected the mixed standards to the LC-MS? Please clearly justify the procedure you have followed for clarity.
5. To investigate the recovery experiment, why S14 was selected?
6. In Table 3: There are various confusing things to be justified. Why different masses of the samples (of course similar and difficult to weigh them practically) were considered? Why you didn't simply consider constant weight of the sample (e.g. 0.5 g as an optimum mass for extraction), spike with known concentration of the analytes, do the extraction and finally evaluate the recovery for all analytes? Besides in column 6, it should be "Recovery (%)" not "Recovry (%).
7. In optimization of the extraction method, I suggest the information presented in Figure 3 to be plotted as "extraction efficiency vs compounds" not as "peak area vs compounds". It is difficult to differentiate and deduce such information in its current form.
8. Table 6: Name column containing sample codes, S1 - S38 and S39 - S75, as "sample number"
9. On preparation of standard solution, (Line 201 - 208) indicate as 80% methanol in water was utilized as a solvent.
10. Line 210: Please rephrase the sentence starting with "0.5 g ...". Starting with numbers is not recommended in scientific writing. Besides, correct "0.5g" to "0.5 g" as space should be let between numbers and units except for percent (X%).
11. Line 212 - 214: The procedure described here need justification. What do you mean by "... made up for lost weight by adding 80% methanol"? How weight loss was observed and how exactly it was compensated?
12. Rephrase Line 215 - 216. It is difficult to understand in its current form.
13. Line 229 - 230: "oC" not "C". Please correct such issues throughout the manuscript, carefully.
14. Overall, significant digit issues should be fixed while presenting the data. The way numerical data were presented in the Tables and the main body are not in scientific way in its current form. Authors need to work, seriously, on this issue.
Round 2
Reviewer 1 Report
The article is now worthy of publication in Molecules.
Author Response
Dear reviewer:
Thank you for your suggestions. we have revised it according to the previous review report.
Reviewer 3 Report
Most of my suggestions forwarded during my first review were addressed and would like to congratulate the authors in this regard. The revision is nice and seems appropriate to readers. However, there is a serious issue yet addressed by the authors:
During my previous review report, I have suggested the authors to re-plot the information presented in optimization of the extraction method as "extraction efficiency" vs "parameters", not as peak "area" vs "compounds" since extraction efficiency will give better information in identifying the optimum conditions. In the revised version, the authors just renamed the y-axis "area" as "extraction efficiency" without calculating the obtained extraction efficiency. Therefore, I suggest to re-plot the figures as "peak area or extraction efficiency" vs "parameters" as follows so that the optimum values can be easily identified.
Fig. 4a, re-plot as "area" vs "solvent type/(%methanol)"
Fig. 4b, re-plot as "area" vs "solvent volume (mL)"
Fig. 4c, re-plot as "area" vs "extraction time (min)"
